# Indoor Visual Exploration with Multi-Rotor Aerial Robotic Vehicles

**DOI:** 10.3390/s22145194

**Published:** 2022-07-11

**Authors:** Panagiotis Rousseas, George C. Karras, Charalampos P. Bechlioulis, Kostas J. Kyriakopoulos

**Affiliations:** 1Control Systems Laboratory, National Technical University of Athens, 15772 Athens, Greece; prousseas@mail.ntua.gr (P.R.); gkarras@uth.gr (G.C.K.); kkyria@mail.ntua.gr (K.J.K.); 2Department of Informatics and Telecommunications, University of Thessaly, 35100 Lamia, Greece; 3Department of Electrical and Computer Engineering, University of Patras, 26504 Patras, Greece

**Keywords:** unmanned aerial vehicles, multi-rotor aerial vehicles, autonomous navigation, robotic exploration, multi-agent systems

## Abstract

In this work, we develop a reactive algorithm for autonomous exploration of indoor, unknown environments for multiple autonomous multi-rotor robots. The novelty of our approach rests on a two-level control architecture comprised of an Artificial-Harmonic Potential Field (AHPF) for navigation and a low-level tracking controller. Owing to the AHPF properties, the field is provably safe while guaranteeing workspace exploration. At the same time, the low-level controller ensures safe tracking of the field through velocity commands to the drone’s attitude controller, which handles the challenging non-linear dynamics. This architecture leads to a robust framework for autonomous exploration, which is extended to a multi-agent approach for collaborative navigation. The integration of approximate techniques for AHPF acquisition further improves the computational complexity of the proposed solution. The control scheme and the technical results are validated through high-fidelity simulations, where all aspects, from sensing and dynamics to control, are incorporated, demonstrating the capacity of our method in successfully tackling the multi-agent exploration task.

## 1. Introduction

In recent years, the field of Robotics has gone through significant advancements in both software as well as hardware. One of the most notable subsets of the field is Unmanned Aerial Vehicles (UAVs). The rise in computing power, advances in energy storage technologies and the development of sophisticated control algorithms have resulted in UAVs being widely employed in research and industry alike. The rapid adoption of such platforms is well-motivated, owing to the versatility and recent accessibility of the platforms, which renders them suitable for a wide range of applications. In this work, we aim to utilize the capabilities of a fleet of multi-rotor autonomous aerial vehicles (drones) for the exploration of indoor spaces. While a single platform can accomplish such a task, it is evident how employing multiple agents results in a decrease in the overall exploration time, as well as overcomes possible limitations due to, e.g., limited battery life. In order to successfully accomplish this task, already existing, robust and proven technologies for visual sensing of the drone’s environment are implemented. The path planning aspect will be tackled through a provably-correct custom framework based on Harmonic Artificial Potential Field (AHPFs). Finally, the control framework is completed through a low-level, custom reactive field tracker, which ensures that the robots track the field safely without sacrificing the full exploration guarantees provided by the AHPF-produced velocity field. Our framework is built around the Robot Operating System (ROS) [1], which enables sensing, communication and actuation of robotic systems. Owing to the aforementioned features, our algorithm exhibits superior performance with respect to smoothness and computational complexity, while also being applicable in realistic scenarios, as validated in a high-fidelity simulation environment.

The sequel is organized as follows: Section 2 concentrates on related work to all key aspects of the proposed scheme. Section 3 formulates the treated problem mathematically, with the proposed solution being extensively discussed in Section 4. We conclude our paper with Section 6 where the limitations of our approach are highlighted along with possible solutions, followed by the Conclusions Section 7, where future prospects are also discussed.

## 2. Related Work

### 2.1. State-of-the-Art on Autonomous Robot Exploration

The problem of autonomous exploration can be summarized as formulating methods for maximizing the information gained over a specific, initially unknown area in which one or more robots operate. Search and rescue reconnaissance missions are a typical example where, due to possible hazardous environments, e.g., after an environmental disaster, robots can be employed to access remote or hazardous areas that are inaccessible to humans [2]. A common approach, termed Frontier-based exploration, consists of defining (multiple) frontiers as regions on the boundary of the so-far explored space that are free of obstacles and/or obstructions. Thus, these frontiers can be employed to provide feasible explorative paths for the robots. The latter are guided towards the aforementioned frontiers while at the same time building a map of the environment until no such frontiers remain [3,4]. There exist various tools in the literature for robots to associate the predicted information gain with the overall exploration progress such as the reduction in entropy of a Rao–Blackwellized Particle Filter [5], Shannon and Renyi entropy [6], Cauchy–Schwarz quadratic mutual information [7] or Gaussian Processes and Bayesian Optimization [8]. Some more recent efforts on autonomous exploration expand on classical navigation algorithms, such as [9], where a Rapidly Random exploring Trees (RRT)-based planner for Unmanned Aerial Vehicles (UAV) is introduced. In [10], a Topology–Grid Hybrid Map (TGHM) scheme is formulated for autonomous exploration. Modern approaches, such as Reinforcement Learning (RL), have also been employed to tackle the exploration problem [11].

It is evident that for the exploration task to be successful, safety during navigation, i.e., lack of collisions between the robots and the obstacles located within the operating area, is pivotal. Another class of solutions to the exploration problem that integrates safety into the exploration problem are Artificial Potential Field (APF) methods, which are employed to construct an underlying velocity vector field over the explored area for the autonomous guidance of the robot towards the unexplored areas. While APFs exhibit several advantages, i.e., reactivity and continuity of the method. etc., some potential fields might present local equilibria that may stall the robot in undesired configurations. To alleviate these limitations, a specific class of APF, namely Harmonic Potential Fields (HPFs), are employed owing to their property of exhibiting no local minima by design. These methods can therefore provide an efficient alternative for autonomous guidance during exploration. Towards this direction, HPFs with Dirichlet boundary conditions [12], Voronoi diagrams and distorted HPFs [13] both for 2D and even for 3D spaces [14] have been presented in the literature. However, constructing such harmonic potential fields often presents numerical and computational challenges and are especially susceptible to the curse of dimensionality, scaling poorly as the size and dimensions of the workspace increase. In order to tackle the above problems, several approaches have been developed, such as restricting the necessary relevant computations of the APF on a local window around each robot [15] or multi-grid methods [16]. Nevertheless, notably, in the aforementioned HPF-based frameworks, the APF was computed by applying finite differences, thus requiring the discretization of the entire exploration domain, which results in a significant increase in the computational complexity proportional to the grid density. More importantly, in most cases, the proposed solution is limited to two-dimensional mobile robots with disc-sector sensing (LIDAR-like sensors).

### 2.2. State-of-the-Art on Multi-Robot Exploration

Fixed-base station multi-agent exploration has been extensively investigated in the literature through a variety of approaches [17,18,19,20,21]; however, communication constraints in such centralized frameworks have not been explicitly considered. Nevertheless, since communication constraints are a persistent aspect of real-world applications, which can hinder the efficacy and implementability of any method, there has recently been a focus on treating multi-agent-related problems under such constraints. The latter have been commonly treated through the notion of connectivity, which concerns the communication ability between robots and the fixed station. Consequently, a widely employed approach consists of maintaining the connectivity between the robots and the base station, directly or in a multi-hop manner, continuously during the execution of any related task. This constraint is especially important in the context of search and rescue applications, where valuable real-time information, such as image streaming, is pivotal [22,23,24]. Other approaches treat communication of acquired information to the base station periodically, in a discontinuous re-connection fashion, which can be implemented either synchronously [25] or asynchronously [26]. In the latter case, periodic connectivity may not be enforced as a hard constraint; for example, in the role-based distributed strategy [27], robots are free to explore the unknown environment with no communication restrictions. In order, however, to communicate relevant information, specifically selected rendezvous points are selected, where asynchronous updates of the relevant robot maps of the environment to the base station take place via communication relays. At the same time, stronger forms of asynchronous connectivity, e.g., line-of-sight communication [28], or a small set of behavioral and message exchanges between robots and dropped beacons [29], have also been investigated. Finally, communication constraints can also be defined with the goal of ensuring global connectivity only at the deployment locations of the robots through recurrent connectivity strategies. This results in enforcing recurrent connectivity upon the collection of new data by each robot [30,31,32]. The motivation behind such approaches mainly rests on the fact that robots can be disconnected for arbitrarily long periods of time while establishing a connection only when new information has been collected.

### 2.3. Proposed Method

Within the existing paradigm, our method aims at constructing a provably correct exploration scheme that employs the capabilities of a fleet of drones to explore an indoor environment in finite time in a safe manner. In order to take advantage of the multiple drones, no explicit communication network is assumed, and each drone can communicate with others wirelessly if and when visual identification between drones is established. This decentralized approach, while not explicitly ensuring optimality during the exploration task, results in a computationally attractive and easily implementable approach in practice, as it only necessitates an adequately equipped drone fleet.

Additionally, in order to formulate a computationally attractive method for real-time octomap building, we decouple the 3D mapping aspect from the navigation task; that is, the navigation framework relies solely on a computationally efficient 2D sector-based scheme. Conversely, the on-board visual information is employed to build a 3D representation of the workspace online. This results in a more robust navigation framework, owing to its simple workspace representation that is more attractive in the context of the on-board computing power of modern multi-rotor platforms. By guaranteeing full workspace exploration, the mapping aspect is also ensured to provide reliable results.

To summarize, our contributions are:
Development of an AHPF-based exploration algorithm for a multi-rotor platform,Extending the above scheme to the multi-robot exploration problem,Integrating the aforementioned exploration framework with a scheme for single-agent visual map-building of unknown workspaces, combined with an inter-agent information exchange aspect.

The goal is, therefore, to provide a computationally tractable solution for indoor exploration of multi-agent, multi-rotor platforms.

## 3. Problem Formulation

Consider a three-dimensional, simply-connected, static workspace $W\in R^{3}$. In this case, the topological property of “simple connectivity” dictates that there are no floating obstacles inside the workspace. Further consider a set of drones, I={1,2,⋯,I}, operating within the workspace W, with $p_{i}\in W-\partial W,\;i\in I$ denoting the position of the *i*-th robot. The problem is thus initially formulated as follows: *Assuming that the drones start from arbitrary positions within W, the goal is to explore the whole workspace in finite time, with no collisions with the workspace boundary ∂W or between the drones themselves*.

In order to formulate a solution, we begin by assuming that there exists a superscribed sphere $S_{i}^{2}\subset R^{⊯},\;i\in I$ with radius $R_{i}$ for each drone, such that each drone lies entirely within the *i*-th sphere. Then, let $W_{infl}$ denote the inflated workspace, i.e.: (1)$$\partial W_{infl}=\left\{q|q=z+R_{max}n\left(z\right),\;\forall z\in \partial W\right\},$$
where $R_{max}=\max\left\{R_{i}\right\},i\in I$ denotes the maximum radius of the set of superscribed spheres, and $n\left(z\right):\partial W\mapsto R^{3}$ denotes the inwards-pointing unitary vector at the point *z* on the boundary of the workspace. Notice that by inflating the workspace, the problem is essentially transformed into the navigation of a *point robot* (this step can be easily incorporated into the proposed framework through computationally inexpensive methods). A necessary assumption that follows is that the inflation of the workspace boundary does not hinder the connectivity of the workspace; when taking the maximal radius for each drone’s representation of the workspace (for compatibility purposes during map matching, as it will become apparent in the sequel, a large enough drone might not “fit” through narrow corridors of the inflated workspace. Therefore, we henceforth assume that $W_{infl}$ remains **fully connected**. Nevertheless, this assumption is not that strong, as the existing platforms relevant to indoor applications are sufficiently small. This assumption can also be lifted if all drones are exactly similar, which is highly likely in practice. Further assume that each drone is equipped with a sensor that provides boundary information of the surrounding workspace, i.e.,
(2)$$S\left(p\right)=\left\{q\in W:\left(B(p,r)\wedge L(p,q)\right)\subseteq W\right\}$$
where B(p,r) is the disk with radius *r* centered at *p* and L(p,q) is the linear segment connecting *p* and *q* inclusively. Additionally, we denote by $P_{i}\left(t_{s},t_{e}\right),i\in I$ the *i*-th drone’s path from time $t_{s}$ until $t_{e}$. The explored region along the path $P_{i},i\in I$ is thus defined as: (3)$$E_{i}≜E\left(P_{i}\right)=\cup_{p\in P_{i}}S\left(p\right)\;,i\in I,$$
whose boundary is denoted by $\partial E\left(P_{i}\right)=\partial E_{F}\left(P_{i}\right)\cup \partial E_{O}\left(P_{i}\right)$, where $\partial E_{F}\left(P_{i}\right)\subset W-\partial W$ denotes the free part and $\partial E_{O}\left(P_{i}\right)\subseteq \partial W$ denotes the obstacle-occupied part of the discovered boundary for the *i*-th drone. We additionally define an appropriate parametrization of the explored boundary, i.e., $\sigma_{i}:\left[0,1\right]\mapsto \partial E_{i},\;i\in I$.

### Proposed Sub-Problems

In order to tackle the above problem, we break it down into three sub-problems, namely:Localization and Mapping,Path Planning,Tracking Problem.

These sub-problems can now be addressed separately. Briefly, the Localization and Mapping problem essentially consists of obtaining an accurate representation of the discovered boundary $\partial E_{i},i\in I$ and of the position $p_{i},\;i\in I$ of each robot. Path Planning is addressed in the context of AHPFs, where a vector field suitable for navigation is constructed. If a robot follows the field exactly, then safe exploration can be guaranteed. However, a robot’s dynamics, along with possible kinematic non-holonomic constraints, prohibit exact tracking of the field. Therefore, we treat the tracking problem separately through the formulation of a provably safe, reactive tracking controller.

## 4. Materials and Methods

We begin with an overview of the method. Based upon the existing attitude controllers of multirotor platforms (e.g., the open source ArduPilot software https://ardupilot.org/, accessed on 1 June 2022), we formulated a two-level controller. The latter is based on a high-level navigation field for exploration [33] and a low-level tracking controller based on the unicycle kinematic model [34]. Finally, inter-agent collision avoidance is guaranteed through the manual addition of the visible subset of agents to each agent’s occupancy grid. Our method provides guarantees of safety and full-workspace exploration in finite time owing to previous works [33,35].

### 4.1. Multirotor Kinematics and Dynamics

In this subsection, we introduce the widely employed kinematic and dynamic models of multirotor platforms. Consider the exemplary multirotor robot, as depicted in Figure 1.

Let $B=\left\{\begin{array}{ccc} e_{B_{x}} & e_{B_{y}} & e_{B_{z}} \end{array}\right\}$ denote the body-fixed frame, whose origin coincides with the vehicle’s center of mass. Furthermore, we define a fixed inertial frame $I=\left\{\begin{array}{ccc} e_{I_{x}} & e_{I_{y}} & e_{I_{z}} \end{array}\right\}$, as depicted in Figure 1. The translational and rotational dynamics of the vehicle are described by the Newton–Euler equations [36,37]:(4)$$\dot{p}={}^{I}v={}^{I}R_{B}{}^{B}v$$
(5)$$m{}^{I}\dot{v}={}^{I}R_{B}F$$
(6)$$J\dot{\omega }=M$$
where $p={\left[\begin{array}{ccc} p_{x} & p_{y} & p_{z} \end{array}\right]}^{T}$, ${}^{I}v={\left[\begin{array}{ccc} v_{x} & v_{y} & v_{z} \end{array}\right]}^{T}$ denote the position and the linear velocity of the drone with respect to the inertial frame I, ${}^{B}v={\left[\begin{array}{ccc} u & v & w \end{array}\right]}^{T}$ is the linear velocity vector with respect to the body frame B, *m* denotes the mass, ${}^{I}R_{B}$ denotes the rotation matrix used to perform rotations from B to I, J denotes the inertia matrix, and ω denotes the angular velocity of the robot with respect to the body frame B.

The external forces and torques that are applied to the vehicle are split into the following terms:(7)$$F=F_{M}+F_{d}+F_{g}$$
(8)$$M=M_{M}+M_{d}$$
where:$F_{d}=C_{d}{}^{B}R_{I}\|{}^{I}v\|{}^{I}v$ denotes the drag forces and $C_{d}$ the drag coefficient matrix;$F_{g}=m{}^{B}R_{I}{\left[\begin{array}{ccc} 0 & 0 & -g \end{array}\right]}^{T}$ denotes the gravitational force, where *g* is the gravitational constant;$F_{M}={\left[\begin{array}{ccc} 0 & 0 & T \end{array}\right]}^{T}$ denotes the total thrust generated by the motors;$M_{M}={\left[\begin{array}{ccc} \tau_{x} & \tau_{y} & \tau_{z} \end{array}\right]}^{T}$ denotes the torque produced by the motors;$M_{d}=C_{m}\|\omega \|\omega$ are the drag moments with $C_{m}$ denoting the drag moment coefficient matrix;

The total input thrust and moment applied to the drone depends heavily on the vehicle’s geometrical characteristics and specifically on the number *N* of motors and the airframe configuration. In line with momentum theory, the thrust force $T_{i}$ and the drag moment $\tau_{i}$ that are produced by the drone’s propellers are assumed to be proportional to the square of the rotor’s angular velocity, i.e.,
(9)$$T_{i}=C_{T}{\omega_{i}}^{2}$$
(10)$$\tau_{i}=C_{\tau }{\omega_{i}}^{2}$$
where i=1,…,N and $C_{T}$, $C_{\tau }$ denote the thrust and drag coefficients correspondingly.

For the quadrotor, used in simulation scenarios, the total thrust and moments are computed by the following control allocation matrix:(11)$$\left[\begin{array}{c} T\\ \tau_{x}\\ \tau_{y}\\ \tau_{z} \end{array}\right]=\left[\begin{array}{cccc} C_{T} & C_{T} & C_{T} & C_{T}\\ -C_{T}l_{x} & C_{T}l_{x} & C_{T}l_{x} & -C_{T}l_{x}\\ C_{T}l_{y} & -C_{T}l_{y} & C_{T}l_{y} & -C_{T}l_{y}\\ C_{\tau } & C_{\tau } & -C_{\tau } & -C_{\tau } \end{array}\right]\left[\begin{array}{c} {\omega_{1}}^{2}\\ {\omega_{2}}^{2}\\ {\omega_{3}}^{2}\\ {\omega_{4}}^{2} \end{array}\right]$$
with $l_{x}$, $l_{y}$ denoting the distance of each motor with respect to the center of mass.

### 4.2. Autopilot and On-Board Sensors

Modern multirotor platforms operate through the application of a low-level controller, which is a cascaded PID control structure consisting of an outer position loop and an inner attitude one. More specifically, the outer position loop is responsible for converting the reference position $p_{d}$, velocity ${}^{I}v_{d}$ (or ${}^{B}v_{d}$) and heading $\psi_{d}$ of the vehicle to target orientation (roll $\phi_{d}$, pitch $\theta_{d}$, yaw $\psi_{d}$) and throttle. The inner attitude controller then translates the aforementioned orientation and throttle commands to motor Pulse Width Modulation (PWM) values. The state feedback is achieved by fusing sensor measurements, such as data by GPS, compass and IMU, using an Extended Kalman Filter. An overview of the control architecture is shown in Figure 2.

### 4.3. Localization and Mapping

In order to perform Localization and Mapping, robust existing SLAM tools will be employed. More specifically, we will employ the Hector SLAM method, as presented in [38]. This framework can be readily integrated through the available, open-source ROS package and provides odometry data for an estimate of a robot’s position, along with a built-in Occupancy Grid Map (OCG), both of which are utilized for path planning and feedback control of the drones. Briefly, SLAM methods utilize sensor information (in Hector SLAM’s case, a LIDAR sensor), which is processed in order to extract an estimate of the position of the sensor through matching sequentially acquired measurements. OCGs, on the other hand, consist of a 2D representation of a workspace and take the form of a matrix. Given an inertial frame of reference (e.g., the drone’s take-off position), the surrounding (unknown) space is decomposed into squares (cells). The position (index pair) of each component of the aforementioned matrix corresponds to a cell inside the workspace, while the value of each component represents the probability that the respective cell is occupied by an obstacle, given the currently available information to the drone. Therefore, having such a representation, obtaining the sensed boundary ∂E can be directly tackled through well-known boundary extraction imaging techniques [39,40], applied over the OCG.

### 4.4. Path Planning

#### 4.4.1. Velocity Field

In this section, we treat the path planning problem for a point-robot (as discussed in Section 3). In order to achieve provably complete path planning for each robot, we employ the AHPF-based method presented in [33]. Briefly, the method designs a reactive vector field for the single integrator dynamics, i.e., given the problem formulation of Section 3, along with the following dynamics: (12)$${\dot{p}}_{i}=u_{i},\;p_{i}\in W,\;i\in I,$$
the authors in [33] provide a reactive vector field, which is modified herein to fit the multi-agent framework: (13)$$u_{i}=u_{i}\left(p_{i}\right)=-K_{u}^{i}S_{R_{1}}\left(d(p_{i};E_{i};p_{I}^{i})\right)\nabla_{p}\phi \left(p_{i};{\hat{k}}_{i}\right),\;i\in I,$$
where $K_{u}\in R$ is a constant gain, the set: (14)$$p_{I}^{i}=\left\{p_{j}|j\subset I-\left\{i\right\}\;\mathrm{and}\;p_{j}\in S\left(p_{i}\right)\right\},$$
for the drone *i* denotes all other drones that lie inside its sensor radius, $S_{R_{1}}:R\mapsto R$ is a scaling function to ensure safety and will be defined in the sequel, and finally, $d(p_{i};E_{i};p_{I}^{i})$ is a distance function: (15)$$d\left(p_{i};E_{i};p_{I}^{i}\right)=\min\left\{\min_{z\in \partial E}\{∥p_{i}-z∥\},\min_{z\in p_{I}^{i}}\{∥p_{i}-z∥-D_{min}^{i}\}\right\},$$
which outputs the minimum distance between the already sensed boundary by the *i*-th drone and its distance from the closest of the other drones. The quantity $D_{min}^{i}>0$ is a parameter that determines the least permissible distance between two drones and is chosen as $D_{min}^{i}<R_{i},\;i\in I$. Furthermore, we define the function: (16)$$S_{a}\left(⋆\right)=\left\{\begin{array}{cc} 0 & ,⋆<0\\ 3{\left(\frac{⋆}{a}\right)}^{2}-2{\left(\frac{⋆}{a}\right)}^{3} & ,0\leq ⋆\leq a\\ 1 & ,⋆>a \end{array}\right.$$

The velocity control law (Equation 13) is further derived from the gradient of the reference potential $\phi (p;\hat{k}):W\mapsto R$, which is determined by a set of parameters $\hat{k}$, which will be defined in the following sub-section. The potential $\phi_{i},\;i\in I$ of the *i*-th drone is designed to satisfy the Laplace equation: (17)$$\nabla_{p}^{2}\phi_{i}\left(p_{i};{\hat{k}}_{i}\right)=0,\forall p_{i}\in W,\;i\in I,$$
with Neumann boundary conditions: (18)$$\frac{\partial \phi_{i}}{\partial p}=k\left(\sigma_{i}^{-1}\left(q\right),t\right),\;\forall q\in \partial E_{i},\;i\in I,$$
where $\sigma^{-1}\left(q\right)$ is the inverse function of the parametrization of the boundary introduced in Section 3. Effectively, the function $k\left(\sigma^{-1}\left(q\right),t\right)$ dictates whether a point on the boundary q∈∂E is attractive (k<0) or repulsive (k>0). In order for the Neumann boundary problem to admit a solution, the boundary conditions should satisfy the compatibility constraint: (19)$$\int_{\partial E}kds=0.$$

We, henceforth, refer to such a set of boundary conditions as *compatible*. The reason why we employ such a potential for navigation is that such solutions lack local equilibria inside the domain over which (Equation 17) is solved. This property holds according to the minimum–maximum principle, which dictates that solutions to the Laplace equation exhibit minimal and maximal values only at the boundary of their domain. The boundary condition (Equation 18) further ensures that the resulting gradient vector field is safe, i.e., the resulting velocities point inwards at the boundary.

#### 4.4.2. Boundary Discretization

It is evident that the Laplace problem (Equation 17), along with the boundary conditions (Equation 18), necessitates the definition of the function k(·), which would result in a problem with an infinite-dimensional boundary condition, as (Equation 18) is defined over a continuous set. In order to handle the boundary conditions numerically, we employ the following parametrization of *k* over a **finite** number of control points $\hat{k}$, i.e.,
(20)$$k_{i}≜k\left(\sigma_{i}^{-1}\left(q\right),t,{\hat{k}}_{i}\right),\;{\hat{k}}_{i}={\left[k_{i}^{1},\cdots ,k_{i}^{N_{C_{p}}}\right]}^{T},\;N_{C_{p}}\in N,\;i\in I,$$
where each $k_{i}^{j},j\in \left\{1,\cdots ,N_{C_{p}}\right\}$ corresponds to points $q_{j}$ placed on the observed boundary. For any boundary point $q_{j}$, ${\hat{k}}_{i}^{j}$ corresponds to the exact value of the boundary condition (Equation 18) for the *i*-th drone, while the same value for any point $q\neq q_{j}$, $k\left(\sigma_{i}^{-1}\left(q\right),t,{\hat{k}}_{i}\right)$ is chosen as a linear combination of nearby control point values. The discretization scheme is illustrated in Figure 3. In addition, we equip the specified boundary conditions $\hat{k}$ with appropriately designed adaptive laws, such that no additional (locally) stable equilibria appear in $\mathrm{int}\left(\partial E\right)$.

These values are tuned according to the adaptive law: (21)$$\dot{\hat{k}}=c\mu \left({\hat{k}}_{t}-\hat{k}\right)+b_{e}\left({\hat{k}}_{t}-\hat{k}\right),$$(the index *i* corresponding to each drone is omitted for the sake of clarity) where $c,\mu ,b_{e},{\hat{k}}_{t}$ are provided in [33] and are omitted here for the sake of brevity. Briefly, the values ${\hat{k}}_{t}$ operate as reference values for the boundary conditions. It is evident that over time, (Equation 21) results in $\hat{k}$ converging to the reference values. Additionally, *c* renders the first term of (Equation 21) zero when the robot is near a critical point of the field ϕ while $b_{e}$ ensures that the trajectories of the robot remain safe for all time. Finally, μ is employed to modify the convergence rate.

#### 4.4.3. Fast Multipole Boundary Elements

In order to obtain a solution to the Laplace Equation (Equation 17) given the boundary conditions (Equation 18) dictated by the function *k* of the preceding sub-section, we employ a Boundary Element Method (BEM) and, more specifically, the Fast Multipole Boundary Elements Method (FBEM), as employed in [33]. Briefly, in FMBEM, the computational cost of the conventional BEM (which is $O(n^{2})$, where *n* is the number of boundary elements used for approximating the domain’s boundary) is significantly reduced through the computation of approximate solutions for the corresponding boundary value problem, with a specified error. In order to accomplish this, a quadtree decomposition is employed in order to obtain a hierarchical subdivision of the elements that describe the boundary of the solution domain. Subsequently, multipole expansion is employed to approximate the interaction among elements (with respect to the Laplace equation). Therefore, an analytical computation of every pairwise element interaction is avoided, at the expense, however, of an approximation error, which is nevertheless bounded. This process results in a reduction in the computational complexity down to O(n). Finally, in FMBEM, there is no need to store the dense and non-symmetric matrix that is part of the conventional BEM if an iterative solver is employed, e.g., GMRES [41], which consequently also results in O(n) memory requirements.

#### 4.4.4. A Brief Discussion about the Algorithm

In this subsection, we briefly explain how the FBEM solution to the problem (Equation 17) and (Equation 18) is employed along with (Equation 21) to provides a safe navigation field suitable for exploration. As in [33], a solution at time T>0 to the problems (Equation 17) and (Equation 18) is obtained by applying the FBEM method, along with the boundary conditions that stem from the parameters’ values at that time, i.e., $\hat{k}\left(T\right)$. Once such a solution is obtained, the linear dependence of the potential on these parameters, along with the adaptive control law (Equation 21), provide a safe field for exploration, as proven by the technical results in [33]. As each drone explores the workspace, the Laplace problem is updated according to newly-discovered parts of the boundary, and successive solutions are obtained. In the following sub-section, we provide the technical results of [33] that prove the asserted claims.

#### 4.4.5. Technical Results

In this subsection, we include some of the technical results presented in [33] for completeness while omitting the relevant proofs for brevity. For details, we direct the reader to [33].

**Proposition** **1.**
*Assuming that $\hat{k}$ is compatible at t=0, the adaptive law *(Equation 21)* guarantees that $\hat{k}$ will remain compatible for all time.*


**Proposition** **2.**
*If the potential $\phi (p,\hat{k})$ is unsafe, the adaptive law *(Equation 21)* guarantees that it will become safe in a finite amount of time.*


**Proposition** **3.**
*All equilibria of the dynamic system:*

(22)
$${\dot{z}}_{i}={\left[{\dot{p}}_{i}^{T},{\dot{\hat{k}}}_{i}^{T}\right]}^{T}$$

*located in int(E) are unstable.*


**Proposition** **4.**(Proposition (8) in [33])**.** (*The proposed control protocol *(Equation 13)* and *(Equation 21)* ensures that the trajectories of ${\dot{p}}_{i}=u_{i},\;i\in I$ are safe.*

The above propositions ensure that the proposed velocity field has the necessary properties that will result in safe exploration of the whole workspace for each drone in finite time. Additionally, in (Equation 13), we propose a modification with respect to [33], which includes the distances between drones that lie inside the sensor radius of each other and results in safety between drones. In order to guarantee this additional type of safety, an additional feature is proposed; namely, within the OCG of each drone, we place a “fictional” obstacle, equivalent to the superscribing sphere $S_{i}^{2}$ that was discussed in Section 3.

### 4.5. Tracking Controller

While we have introduced a provably complete navigation field for each drone, we need to ensure that, despite the inability of a drone to follow the single integrator dynamics exactly (Equation 12), safety is not violated. We assume that there exists an attitude controller over the drone dynamics that takes reference velocities as inputs and results in an adequate response by the drone. Such controllers are in widespread use in the industry and are standard in any available drone platform [42,43]. Additionally, we need to ensure that the drone’s visual system is also pointing towards the direction of the motion, which, besides ensuring proper sensing, results in a better response with respect to the commanded velocities, as the drone is not commanded to alter its direction of motion in a sharp, non-smooth fashion. If the drone’s low-level tracking controller was commanded to change its direction in such a fashion, this would result in a delayed response (owing to the drone’s dynamics and its momentum), which might render its motion unsafe. To this end, we first introduce the yaw angle of the drone $\theta \in S^{1}$ (where $S^{1}$ is the set of 1-dimensional rotations), which can be obtained with respect to an inertial frame of reference {I} as $\theta =\mathrm{atan}2{(}^{I}p_{x}{,}^{I}p_{y})$, where ${[}^{I}p_{x}{,}^{I}p_{y}]$ denotes the *x* and *y* axes of the drone’s own frame of reference expressed in {I}. Then, we propose the following reactive control law: (23)$$\dot{\theta }=-K_{\theta }\left(1-\cos(\alpha )\right)\sin\left(\alpha \right),$$
where α is the relative error angle, i.e.,
(24)$$\alpha =\theta -\theta_{\mathrm{ref}},$$
where $\theta_{\mathrm{ref}}$ is the reference angle of the underlying velocity field, i.e.,
(25)$$\theta_{\mathrm{ref}}=\mathrm{atan}2{(}^{I}u_{x}\left(p\right){,}^{I}u_{y}\left(p\right)),\;u\left(p\right)={\left[{}^{I}u_{x}\left(p\right){,}^{I}u_{y}\left(p\right)\right]}^{T}.$$We prove how this control law stabilizes the reference angle at $\alpha =0\Leftrightarrow \theta =\theta_{\mathrm{ref}}$:

**Proposition** **5.**
*The controller *(Equation 23)* results in zero relative error angle over time for a constant reference angle $\theta_{\mathrm{ref}}$.*


**Proof.** Consider the Lyapunov Candidate function $V\left(\alpha \right)={\left(1-\cos(\alpha )\right)}^{2}$. This funtion is evidently positive definite $\forall \alpha \in S^{1}$, with
(26)$$V\left(\alpha \right)=0\Leftrightarrow \alpha =0\Leftrightarrow \theta =\theta_{\mathrm{ref}}.$$Additionally, its derivative with respect to time is:
(27)$$\begin{array}{cc} \dot{V} & =+2\left(1-\cos\left(\alpha \right)\right)\sin\left(\alpha \right)\left(\dot{\theta }-{\dot{\theta }}_{\mathrm{ref}}\right)\\ \; & =-2K_{\theta }{\left(1-\cos\left(\alpha \right)\right)}^{2}\sin^{2}\left(\alpha \right)-2\left(1-\cos\left(\alpha \right)\right)\sin\left(\alpha \right){\dot{\theta }}_{\mathrm{ref}} \end{array}$$
which for ${\dot{\theta }}_{\mathrm{ref}}=0$ is negative for α≠0, thus concluding the proof. □

However, it is evident that ${\dot{\theta }}_{\mathrm{ref}}\neq 0$ as the drone moves inside the workspace, which changes the vector of the reference velocity and, therefore, its reference angle as well. To remedy this, we propose the following velocity controller: (28)$$u_{\mathrm{ref},i}\left(p\right)=K_{u}\left[\gamma \sigma_{p}\left(\frac{d(p_{i};E_{i};p_{I}^{i})}{\epsilon_{p}}\right)+\left(1-\gamma \right)\sigma_{u}\left(\frac{\langle {\hat{e}}_{n}^{i},u\left(p_{i}\right)\rangle \left(-\langle {\hat{e}}_{n}^{i},\hat{n}\left(p_{i}\right)\rangle \right)}{\epsilon_{u}+|\langle -{\hat{e}}_{n}^{i},u\left(p_{i}\right)\rangle \left|\right|\langle {\hat{e}}_{n}^{i},\hat{n}\rangle |}\right)\right]u\left(p_{i}\right),$$
where γ∈(0,1), $\sigma_{p},\sigma_{u}:R^{+}\mapsto \left[0,1\right]$ are smooth and monotonic bump functions such that $\sigma_{p}\left(0\right)=0,\sigma_{p}\left(\infty \right)=1$. The quantity ${\hat{e}}_{n}^{i}$ denotes the unitary vector along the direction of the *i*-th drone’s heading (forward-looking direction), $\hat{n}\left(p\right)$ denotes the unitary, inwards-pointing vector at the closest point of boundary to the drone’s position *p*, and $\langle \cdot ,\cdot \rangle$ denotes the classical inner product between two vectors, while u(p) denotes the reference velocity field presented in Section 4.4. More specifically, we define the normal vector at the *i*-th drone point $p_{i}$: (29)$$\hat{n}\left(p_{i}\right)=n\left(z\left(p_{i}\right)\right),\;z\left(p_{i}\right)=\arg\min_{z\in \partial E_{i}\cup p_{I}^{i}}\{∥p_{i}-z∥\},$$
where evidently $n\left(z\right),\;z\in \partial E_{i}\cup p_{I}^{i}$ denotes the normal vector to the boundary of the so-far explored region or the closest of the rest of the robots (if it lies within the *i*-th robot’s sensor radius). The first term in (Equation 28) becomes zero as the drone approaches the boundary of another drone (where $d(p;E_{i};p_{I}^{i})\to 0$). The second term approaches zero as:The vectors ${\hat{e}}_{n},u\left(p\right)$ point to different directions, andThe vectors ${\hat{e}}_{n},\hat{n}$ point to different directions.

The first condition means that the velocity of the drone decreases if the robot is misaligned with respect to the velocity field, while the second condition decreases the velocity if the drone is pointing outwards at the boundary. Additionally, this implies that if the functions $\sigma_{p},\sigma_{u}$, along with the tuning parameters $\gamma ,\epsilon_{p},\epsilon_{u}$, are chosen appropriately, they provide an adequate bound to $\dot{\theta }$, which renders the controller (Equation 23) asymptotically stable for ${\dot{\theta }}_{\mathrm{ref}}\neq 0$. This results in safe and successful navigation, regardless of the drone’s dynamics, if the drone’s own attitude controller can perform adequately. The efficacy of the above-proposed control protocol in successfully completing the multi-drone exploration task is demonstrated in rigorous, high-fidelity software-in-the-loop (SITL) simulations.

### 4.6. Octopmap Building

Since the navigation aspect is fully addressed through a two-dimensional OCG framework, the visual information is employed independently in order to build a three-dimensional representation of the indoor environment. This is achieved through integrating already existing, robust tools for 3D mapping. The independence of the navigation and 3D mapping modules results in a computationally more attractive scheme. We demonstrate the efficacy of the proposed method in building a complete map in the results section of our work in a realistic, synthetic simulation environment.

### 4.7. Drone Communication and Map Merging

In order to exploit the full potential of the multi-drone exploration framework, a method for establishing communication between the drones to exchange the robots’ maps needs to be formulated. Subsequently, each robot should process the exchanged map information and merge the latter with its own map in order to take advantage of the exploration of the rest of the drone fleet. To accomplish this, we formulate a decentralized, aperiodic communication strategy. In order to provide a simple solution that negates the need for a centralized station, we assume that during the exploration process, a number of C∈N drones, denoted by $\left\{i_{1},i_{2},\cdots ,i_{C}\right\}≜C\subset I$, lie in the visual field of one or more of the aforementioned drones. We formulate the communication scheme for one drone, denoted by i∈I, and employ the same strategy for any other drone. In order to verify the presence of the set of drones $C_{i}\subset I-\left\{i\right\}$ for the *i*-th drone within its vicinity, we use visual means. This choice further motivates the Lyapunov-based controller for each drone’s heading angle θ (Equation 23), as it is necessary that each drone’s visual system is appropriately oriented towards its direction of motion in order to possibly detect any other neighboring drone. We further assume that the presence of two drones within the field of view of any of the latter implies that wireless communication is feasible. Having successfully satisfied the above conditions, the *i*-th drone can send its formed OCG map towards every drone $j\in C_{i}$ in its vicinity.

A pivotal aspect to the success of the scheme is the merging, by the *i*-th drone, of a composed map from the set of neighboring drones $C_{i}$. There exist several methods in the literature that achieve this merging, from [44], where SLAM was extended for a set of multiple robots based on an Extended Kalman Filter (EKF), to [45], which employed a particle filter to achieve multi-robot SLAM. In another approach [46], the author proposes a hybrid method for the formation of a maximum likelihood map along with a Monte-Carlo position estimator. Furthermore, there are off-the-shelf ROS packages available for map-merging, e.g., [47]. In this work, we employ this tool to achieve the map merging for each drone.

## 5. Results

In order to demonstrate the efficacy of the proposed scheme, we have constructed a synthetic simulation scenario where two drones operate within an a priori unknown indoor environment, comprising of planar vertical walls. The environment is depicted in Figure 4. As the drones navigate autonomously, they gather visual data and build an Octomap 3D representation of their environment. Successive snapshots during the exploration process are depicted in Figure 5. We note that both Octomaps, one formed by each drone, are merged and presented in the respective snapshots for illustration purposes. The final Octomap, along with the merged OCG maps after the exploration process has been completed, are presented in Figure 6. Furthermore, the trajectories of the two drones overlayed on top of the final OCG map are depicted in Figure 7. Finally, a video of the experiment can be accessed through the following hyperlink: https://youtu.be/yDP_Zz-P0lE (accessed on 11 June 2022).

## 6. Limitations

The main limitations of this work are related to sensing and tuning. Most multi-rotor platforms are not equipped with an omni-directional Lidar, but rather their sensing is limited over a smaller subset of the full 360° range. This specification might result in the formal assumptions of the method being violated, owing to the drone “exiting” the sensed area over which the AHPF is defined, evidently due to the drone’s blindspots. This issue can be addressed if, prior to initiating exploration, the drone performs a full rotation about the vertical axis of its body frame of reference, thus forming an appropriate “sensed region” around itself and negating the probability of exiting the aforementioned area.

The second limitation is due to the use of Lyapunov/PID tracking controllers. Since these controllers are model-agnostic, some tuning is required to ensure proper tracking of the underlying AHPF. Nevertheless, tuning the respective parameters did not prove to be difficult in practice, especially since a conservative choice of maximum linear velocity can easily result in safe navigation, however, sacrificing total exploration time. An interesting future direction rests on exploring tuning methods for the parameters of the herein proposed controllers.

Another envisioned limitation—although not observed in practice during the evaluation of the proposed scheme—lies in the computational complexity of the method in large workspaces. While the FBEM is advantageous insofar as it scales linearly with respect to the size of the boundary (in contrast to scaling proportionally to the area of the workspace), very large workspaces could still pose an issue as the whole workspace is considered during the solution to the Laplace problem. This could evidently be sidestepped, as parts of the explored workspace could be discarded. This is another interesting future research area where we intend to formulate a polygonal convex decomposition method for reducing the computational load of the proposed scheme in large workspaces.

A final limitation stems from the limited battery life of drones, which is crucial in the context of exploration, as the size of the workspace is not known a priori. This could be addressed through existing approaches, such as [48].

## 7. Conclusions

In this work, we have formulated a method for solving the cooperative exploration task through a multirotor-based scheme. The proposed framework relies on existing robust tools for mapping and localization, while its novelty lies in the application of a Harmonic Potential-based exploration algorithm. Through the technical results, along with the in-simulation validation of the proposed control scheme, our method effectively tackles the multi-agent exploration problem with multi-rotor aerial robots. We, therefore, conclude that the herein posed research problem was successfully solved. Furthermore, our method exhibits some significant advantages. Since the proposed scheme employs the FBEM method, transforming the problem into a boundary problem on a 1D curve, it scales favorably with respect to the size of the workspace (O(n) complexity). This is in contrast to other popular methods, such as other popular planners for navigation in unknown workspaces, such as A${}^{⋆}$ [49], which, at best, scales polynomially or, at worst, exponentially with respect to the size of the workspace. Another advantage of the method is the two-layer approach through a tracking controller over a navigation field. This results in a more robust scheme with respect to the drone’s dynamic behavior, as the internal attitude controller of the multi-rotor platform is employed to tackle the stabilization of its highly non-linear dynamics. Furthermore, our method is modular, which means that it can be easily employed in several robotic platforms (e.g., mobile robots) with minor modifications. This results in a robust and computationally efficient framework for cooperative exploration of indoor areas, as demonstrated in a high-fidelity realistic simulation scenario.

With regards to future work, we intend to improve several aspects of the algorithm. First of all, the parameters of the proposed tracking controller are tuned by hand. It would, however, be possible to formulate tuning laws based on some optimality criterion, e.g., exploration time, explored area coverage, etc., in a multi-agent, game-theoretic approach. Additionally, since computing the field for the whole workspace is unnecessary in several cases, thus adding to the computational complexity of the method with no apparent advantages, we intend on formulating an algorithm for disregarding subsets of the explored workspace that do not "contribute" to the discovery of new parts of the workspace, based on polygonal decomposition. This will effectively bound the number of parameters as in [50].

## Figures and Tables

**Figure 1 sensors-22-05194-f001:**
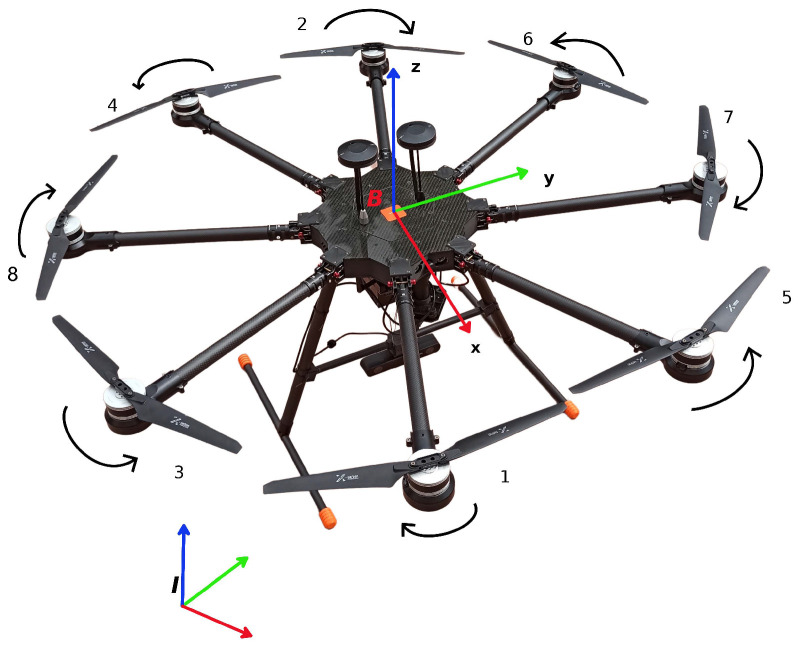
Multirotor’s frames.

**Figure 2 sensors-22-05194-f002:**
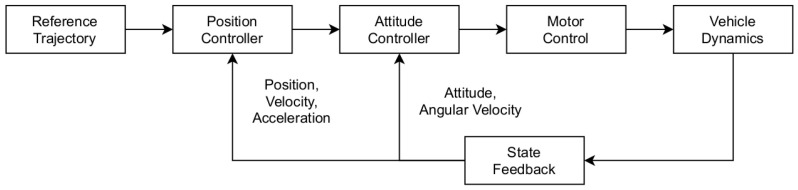
Ardupilot control architecture.

**Figure 3 sensors-22-05194-f003:**
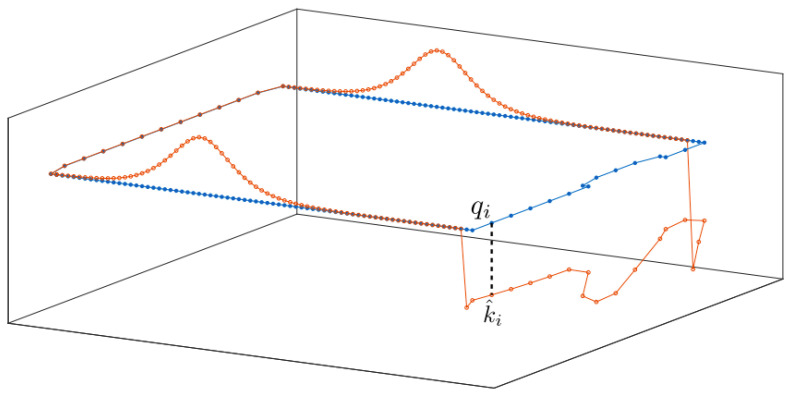
The discretization of boundary values using the control points ${\hat{k}}_{i}$ on the boundary points $q_{i}$. The blue line and dots represent ∂E and the boundary points $q_{i}$, respectively. The red line illustrates *k*, and the control points ${\hat{k}}_{i}$ are shown as red circles.

**Figure 4 sensors-22-05194-f004:**
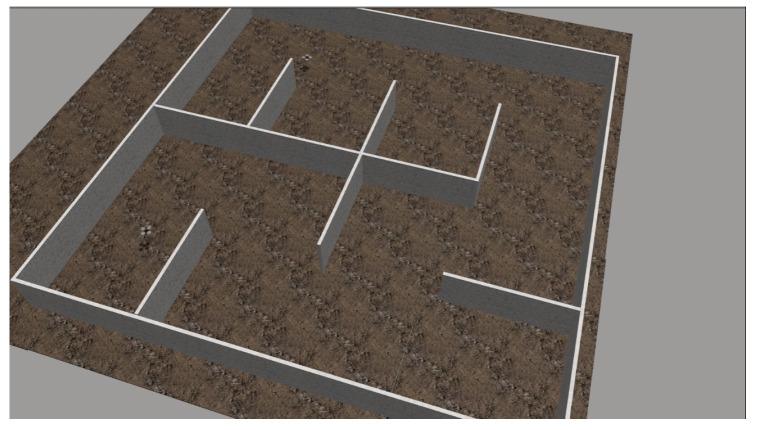
The synthetic simulation environment along with the two autonomous drones.

**Figure 5 sensors-22-05194-f005:**
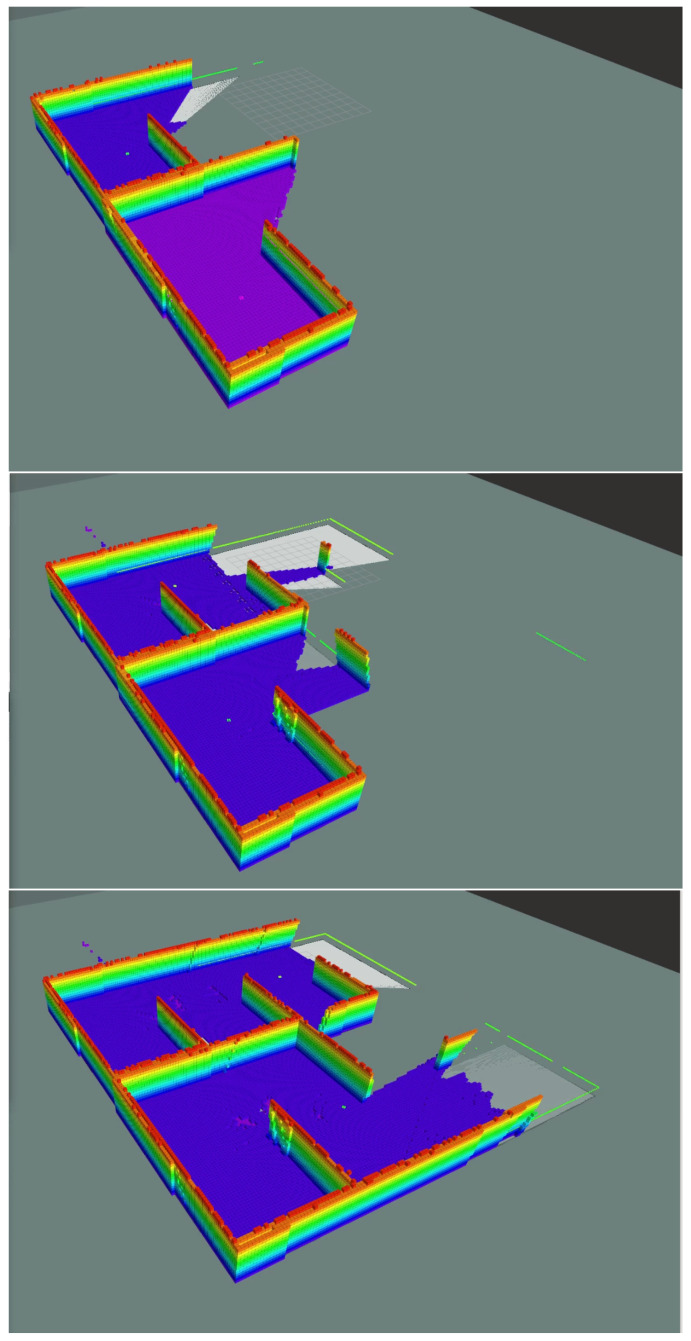
Snapshots of the map-building process during exploration.

**Figure 6 sensors-22-05194-f006:**
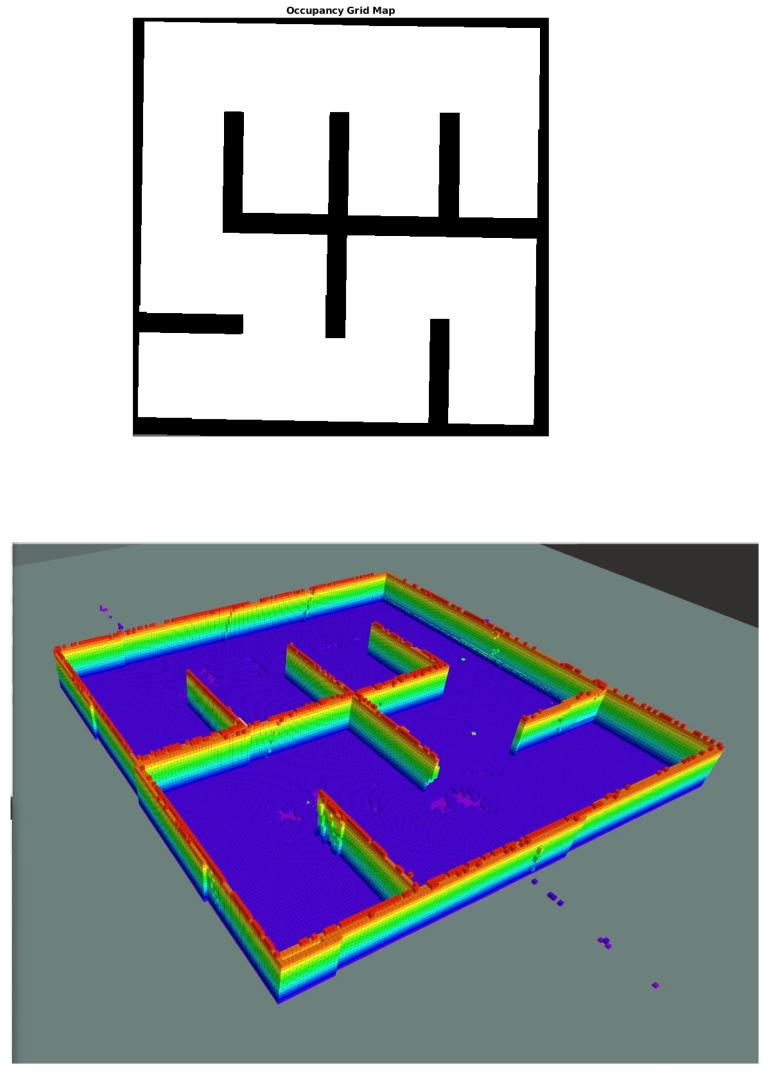
Final results of the exploration/mapping framework. The complete 2D OCG map of the environment (**top**), along with the final 3D Octomap representation of the latter (**bottom**).

**Figure 7 sensors-22-05194-f007:**
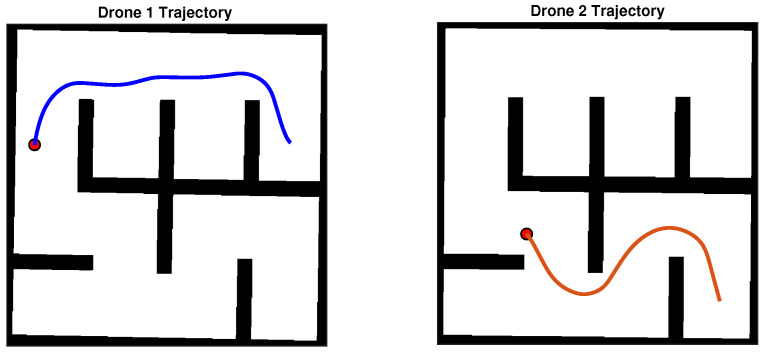
The trajectories that the two multirotor robots executed during the exploration process.

## Data Availability

Not applicable.

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
