# Peer review of "Indoor Visual Exploration with Multi-Rotor Aerial Robotic Vehicles"

_sensors, 2022, doi:10.3390/s22145194_

Round 1
Reviewer 1 Report
Reviewer's summary after reading the manuscript:
In this work, the authors develop an autonomous and reactive exploration algorithm for indoor applications employing multi-rotor aerial robotic vehicles and visual feedback from onboard cameras. The approach is based on harmonic potential fields that guarantee safe navigation avoiding collisions with the workspace boundaries as well as complete exploration of the overall environment. The architecture is extended for multiple aerial vehicles including collision avoidance among them as well as an algorithm for merging the gathered information so that the exploration task is accomplished faster as the number of aerial robotic vehicles scales up. The framework is tested in realistic simulations to verify the theoretical findings. The following are the authors' contributions to the research:
- Development of an AHPF-based exploration algorithm to a multi-rotor platform;
- Extension of the scheme described above to the multi-robot exploration problem;
- Integration of the aforementioned exploration framework with a scheme for single-agent visual map-building of unknown workspaces, combined with an inter-agent information exchange aspect.
Therefore, their objective is to come up with a solution that can be solved with a reasonable amount of computing power for the indoor exploration of multi-agent, multi-rotor platforms.
----------------------------------------
Dear authors, thank you for your manuscript. I enjoyed reading it. Presented are some suggestions to improve it:
(1) Please include a "Limitations" section to describe the difficulties your team encountered and how they overcome them. The readers would gain a lot from this because they might benefit from your extensive experience.
(2) Conclusions are too brief. The final section "Conclusions" needs to be greatly expanded. The authors must answer the Research Question, and also address how the Research Problem is ameliorated in the Conclusions section. The "Directions of Future Research" can also be included in the Conclusions section.
(3) If you want your paper to have a greater impact and a larger audience, you need to explain why and how this piece is distinct from others of a similar kind in the Abstract, the Introduction, and the Conclusion sections.
(4) Please expand your literature review and provide additional references to MDPI journal papers.
(5) MDPI's formatting rules have not yet been followed by all of the references listed. Not all of the cited references' DOIs have been included. When arranging your references, please use, for example, the free Zotero software and select "Multidisciplinary Digital Publishing Institute" as the citation format. There are already 39 references in your paper, and there are likely to be more once you have amended the document.
I appreciate your consideration of these comments.
Thank you.
Author Response
Thanks for your comments. The manuscript has been revised accordingly to comply with them. See the attached file. All modifications have been highlighted in blue color.

Reviewer 2 Report
Please, refer to the attached document. Thanks!

Author Response

(The authors gave the same response as above.)

Round 2
Reviewer 2 Report
Please, refer to the attached document. Thanks!
